# Antioxidant Activity of Beef, Pork and Chicken Burgers before and after Cooking and after In Vitro Intestinal Digestion

**DOI:** 10.3390/foods12224100

**Published:** 2023-11-12

**Authors:** Giulia Grassi, Giambattista Capasso, Andrea Rando, Anna Maria Perna

**Affiliations:** 1Department of Agriculture, Environment and Food, University of Molise, Via De Sanctis 1, 86100 Campobasso, Italy; 2School of Agricultural, Forestry, Food and Environmental Sciences, University of Basilicata, Viale dell’Ateneo Lucano 10, 85100 Potenza, Italy; giambattista.capasso@unibas.it (G.C.); andrea.rando@unibas.it (A.R.); anna.perna@unibas.it (A.M.P.)

**Keywords:** meat, species, antioxidant activity, in vitro gastro-intestinal digestion

## Abstract

The aim of the present work was to evaluate and compare in vitro the antioxidant activity of raw, cooked and cooked–digested pork, beef and chicken burgers. The cooking process influenced the antioxidant capacity of the meat by decreasing the values of ABTS, FRAP and the content of free thiols. Conversely, a positive effect was observed after in vitro gastrointestinal digestion which increased the biological activity of the meat, characterised by greater antioxidant activity. The type of meat influenced the chemical composition and biological capacity of the burgers. In fact, both before and after the cooking process, beef burgers showed higher thiol content and, consequently, a higher oxidative stability of proteins than chicken and pork burgers. In vitro gastrointestinal digestion also improved the nutraceutical quality of beef burgers, which showed higher ABTS values and thiol content than pork burgers, which showed higher FRAP values. This work aims to support the potential of meat constituents as a natural antioxidant component that is essential to counteract the oxidative stress responsible for imbalances in the human organism and several cardiovascular diseases.

## 1. Introduction

In everyday life, food plays a fundamental role in the health of the consumer, who selects and searches for a source that is not only nutritious but also functional [1]. Oxidative stress, i.e., a condition of strong imbalance between free radicals and reactive oxygen species (ROS), has been linked to several pathologies, including cancer, cardiovascular diseases and diabetes [2,3], while the consumption of foods rich in antioxidants, including peptides with high antioxidant capacity, has been correlated with a lower presence of oxidative stress [4,5,6]. The scientific community is strongly interested in the components (antioxidants) linked to the prevention of the harmful effects that free radicals have on the human organism. Currently, many researchers are focused on various foods, such as vegetables, fruit, aromatic herbs, spices and honey [7,8,9], while there is little scientific information on the antioxidant properties of foods of animal origin. Meat is a very rich source of high-quality organic protein and provides the body with all the essential amino acids that are needed in the diet [10]. Furthermore, meat is rich in micronutrients, especially group B vitamins and minerals, such as P, Mg, Co, Zn, Fe and Se; it also contributes to the supply of omega-3 fatty acids and vitamin E [4,5,11]. Meat is characterised by many endogenous enzymatic and non-enzymatic antioxidants and a hydrophilic or liposoluble nature that can minimise rancidity in foods, with the consequence that it is identified as a functional food [12,13]. Carotenoids, for example, are non-enzymatic antioxidants [4], but there are also unsaturated aliphatic compounds (characterised by their orange-pink colour) and vitamins A and E (α-tocopherol, the most lipophilic and active antioxidant) [11,12]. Hydrophilic antioxidants, including bioactive peptides such as carnosine (β-alanyl-l-histidine), anserine and others, are components naturally present in vertebrate animal tissues, especially in skeletal muscle [5,10,14]. Carnosine is characterised by the presence of histidine and plays a specific and beneficial role in various physiological functions, including antioxidant activity [3,15]. Furthermore, meat has endogenous antioxidants, mainly represented by enzymes (catalase, superoxide dismutase and glutathione peroxidase—GPx), which play a defensive role against oxidative stress and free radicals excess [12,16].

All enzymatic and non-enzymatic antioxidant systems counteract the action of pro-oxidants in muscle tissue [14], both in live animals and after slaughter. The composition of endogenous antioxidants can be influenced by several factors including type of meat, breed [15,16] and muscle type. The animal’s diet also plays a significant role in modifying the concentration of antioxidants, pro-oxidants and fatty acids in meat [17]. Meat is generally not considered a source of dietary antioxidants; however, some authors [18] demonstrated that the consumption of beef significantly increased the total antioxidant capacity (TAC) of human serum. Furthermore, oral administration of L-carnosine increased the serum TAC level. Carnosine is the main component with antioxidant activity and anserine has also been reported as an effective hydrophilic antioxidant [18]. Dipeptides’ antioxidant capacity is due to both radical scavenging activities and the chelating properties of metals [19,20]. Bioactive peptides are amino acid sequences that are inactive within the parent protein sequence but are activated following proteolytic processes (enzymatic or chemical hydrolysis), microbial fermentation or gastrointestinal digestion [21]. In addition, some endogenous antioxidants specific to meat, such as dipeptides that contain 2-oxo-imidazole (2-oxo-IDP), were formed, in turn, from histidine or its methylated derivatives, which had a marked antioxidant activity even greater than other dipeptides, and have been observed in different meats (beef, pork and chicken) [22].

Before being consumed, meat requires a cooking treatment, which is responsible for compositional and structural changes that, consequently, influence the total antioxidant capacity of the meat [12,23]. Many authors have demonstrated the negative impact of cooking processes on protein oxidation and the decrease in antioxidant content [14,17]. The decrease in the content of components that perform bioactive activities as thiols is due to the reactions that take place with free radicals or which form intermolecular disulphide bridges, which prevent their maximum expression of activity [24]. Following gastrointestinal digestion, however, their activity increases as the bioactive components remain inactive within the proteins, and only following the enzymatic action of digestion and their release does their activity increase [14,25]. Therefore, meat can be considered a potential food from which to obtain biopeptides following enzymatic hydrolysis, which performs fundamental biological activities such as antihypertensive, antithrombotic, immunomodulatory, antimicrobial and antioxidant actions [12,23,25]. In conclusion, different classes of antioxidants are present in meat which can have different protective effects against oxidation; however, an overall measurement of antioxidant potential can be more important than the concentration of each individual compound [26]. The antioxidant activity, therefore, could be an indicator of beneficial effects for the consumer’s health.

The aim of the present work was to evaluate and compare, in vitro, the antioxidant activity of raw, cooked, and digested–cooked pork, beef and chicken burgers. The choice to use burgers arose from the widespread diffusion of this type of preparation, which is highly appreciated by consumers aged 6 to 25 years [27,28].

## 2. Materials and Methods

### 2.1. Experimental Design

The study was carried out on 18 burgers prepared with meat from three different meats, beef, chicken and pork, purchased from a local supermarket in Potenza (southern Italy). After mincing the meat (1 kg of meat for each type), 6 burgers for each type were prepared, with the addition of salt (26 mg NaCl/100 g of meat). Raw meat samples of each meat were prepared under the same conditions, four for each. For burgers of each meat, fresh rib meat was used for the beef burger (BH), fresh pork loins for the pork burger (PH) and breast and leg meat in a 1:1 ratio for the chicken burger (CH). From each meat preparation, burgers were made using a special shaper with dimensions of 7 cm × 1.5 cm for 100 g of meat. CH, BH and PH were stored at refrigerated temperature (+4 °C) until analysis.

### 2.2. Cooking Process

The cooking process was performed as suggested by Simonetti et al. [12], with some modifications. Six burgers per meat were placed in a convection steam oven (Küppersbusch CPE 110, Küppersbusch Grobküchentechnik GmbH, Gelsenkirchen, Germany) set at 120 °C and equipped with a thermometer placed inside each burger. Once 75 ± 3 °C was reached in the centre of the burger, the cooking process was considered complete (5 min). Subsequently, the meat samples were cooled in an ice bath and stored at −20 °C until analysis. All cooked burgers underwent in vitro gastrointestinal digestion and were analysed for antioxidant activities.

### 2.3. Chemical Composition

In accordance with AOAC [29], the proximate composition was determined. The proteins, total fats, ash and dry matter were determined on raw and cooked burger samples of each meat. Each determination was performed in triplicate.

#### 2.3.1. In Vitro Gastrointestinal Digestions

The method suggested by Simonetti et al. [12] was used to simulate in vitro gastrointestinal digestion. Frozen cooked burgers were homogenised using a laboratory blender and divided into 5 g portions. After, the samples were mixed with 50 mL of double-distilled water and subjected to the simulation of human chewing using a Stomacher (Steward Stomacher 400 Lab Blender, London, UK) for 60 s. The pH was adjusted to 2 (model PHM 92, Radiometer, Copenhagen, Denmark) using 3 M HCl (Sigma-Aldrich, Milan, Italy) and pepsin (Sigma-Aldrich P6887, Milan, Italy) added in a ratio of 1:100 (enzyme/substrate) in order to simulate the gastric phase. The digestion process was subjected to 37 °C for 2 h under continuous stirring, and the enzyme was inactivated by adjusting the pH to 7.2 with 1 M NaHCO_3_ and pancreatin (Sigma-Aldrich P3292, Milan, Italy) in a ratio of 1:50 (enzyme/substrate) at the simulated intestinal phase. Digestion was carried out for another 3 h at 37 °C and, subsequently, the enzymatic activity was interrupted by increasing the temperature to 95 °C for 10 min. The reaction mixture was centrifuged at 5000× *g* for 20 min at 4 °C (CR 412, Jouan, Saint-Herblain, France), filtered with a 0.45 µm cellulose acetate membrane filter (Sigma-Aldrich, Milan, Italy) and stored at −55 °C until analysis. Simultaneously, a blank test was simulated in order to evaluate the possible impact of digestive enzymes on the planned analyses. The mixture was made up of water (instead of the sample) and gastrointestinal juices and enzymes. Furthermore, digestion of the different types of meat in the absence of digestive enzymes was also performed to distinguish the effects due to the presence of enzymes from those caused by the chemical conditions in the test.

#### 2.3.2. Preparation of Raw, Cooked and Digested Burgers for Antioxidant Activity

The preparation of raw and cooked burger extract was carried out as suggested by Perna et al. [23]. Aliquots of meat (2 g), raw, cooked and digested, were homogenised with 6 mL of distilled water using a Polytron (PT-MR 2100, Kinematica AG, Littau, Lucerne, Switzerland) at 13,500 rpm for 15 s. Subsequently, they were placed in an ultrasonic bath (US) (Elma Transsonic 460/H, Singen, Germany) for 10 min at room temperature and centrifuged at 5000× *g* at 4 °C for 20 min. The supernatant was filtered with a 00.45 µm cellulose acetate membrane filter (Sigma-Aldrich, Milan, Italy), and subjected to analysis.

#### 2.3.3. Free Thiol Groups

Simonetti et al. [12] suggested a method to determine the concentration of free thiol groups according to the Ellman method [30]. The supernatant volume (raw, cooked and digested) of 250 µL of each extracted sample was mixed with 2.5 mL of 0.1 M sodium phosphate buffer (containing 1 mM ethylenediaminetetraacetic acid (EDTA), pH 8.0, reaction buffer; Sigma-Aldrich, Milan, Italy) and 50 µL of 5–50-dithiobis (2-nitrobenzoic acid) (DTNB) reagent solution (4 mg in 1 mL of sodium phosphate buffer; Sigma-Aldrich, Milan, Italy). The reaction was left at room temperature for 30 min and the absorbance was read at 412 nm using a 1204 UV–vis spectrophotometer (Shimadzu, Kyoto, Japan). The samples were read against the blank which contained the reaction buffer in place of the sample. The results were expressed as µmolSH/g of meat, considering the molar extinction coefficient of 14,150 M-1 cm^−1^. Each determination and measurement was performed in triplicate.

#### 2.3.4. ABTS Assay

The method suggested by Re et al. [31] was applied to assess the scavenging capacity of raw, cooked and digested–cooked burgers, with some modifications. The ABTS^•+^ radical solution was produced by reacting 10 mL of the ABTS stock solution (7 mM) with 175 μL of potassium persulfate solution (140 mM). The mixture was left in the dark at room temperature for 16 h before use. Before starting the spectrophotometric analysis, the dark ABTS^•+^ solution was diluted with ethanol (96%) to obtain an absorbance of 0.700 ± 0.020 at 734 nm. Then, 2 mL of ABTS^•+^ solution and 100 μL of the extracted samples were placed in a cuvette, and after 30 min the decrease in the absorbance was measured. The reagent blank was prepared by adding 100 μL of ethanol instead of the sample. A calibration curve was made using known concentrations of 6-hydroxy-2,5,7,8-tetramethylchroman-2-carboxylic acid (Trolox) (20, 50, 150, 400, 700 and 1000 μg/mL) as standards, and the results were expressed as μg of Trolox equivalent per g of meat. Each determination and measurement was performed triplicate.

#### 2.3.5. FRAP Assay

The ability to reduce the ferric to ferrous ion of raw, cooked and digested–cooked burger samples was evaluated as suggested by Anoudi et al. [32], with some modifications.

The FRAP reagent was composed using 10 volumes of buffer acetate (0.3 M, pH 3.6), 1 volume of TPTZ (10 mM in 40 mM HCl) and 1 volume of iron chloride (20 mM). The reagent was placed at 37 °C until the end of the analyses. Then, 100 μL of extracted sample and 3 mL of FRAP reagent were placed in cuvettes. The absorbance of each mixture was measured after 30 min of incubation at 37 °C, at 593 nm against the blank. The blank was prepared using 100 µL of acetate buffer in place of the sample. A calibration curve was made using known concentrations of Trolox as the standard, and the results were expressed as μg Trolox equivalents (TE) per g of sample. Each determination and measurement was performed in triplicate.

### 2.4. Statistical Analysis

The data were statistically analysed by two-way analysis of variance (ANOVA) using the general linear model (GLM) at a 95% confidence level (*p* ≤ 0.05) in the SAS 1972 software [33] with interactions according to model:y_ijk_ = μ + α_i +_ βj + (α_i_ × βj) + ε_ijk_
where y_ijk_ is the experimental observation; μ is the mean; α_i_ is the fixed effect of the ith meat (i = beef, chicken, pig); βj is the fixed effect of the jth meat state (j = raw, cooked and digested–cooked); α_i_ × βj is the effect of the interaction between the meat type and the meat state; ε_ijk_ is the random error. Before setting the values, expressed in percentage terms, they were subjected to arcsine transformation [12].

The Tukey post hoc test was used for comparison of means, and differences were considered significant (*p* < 0.05). Results are presented as mean ± standard deviation (±SD).

## 3. Results and Discussion

### 3.1. Chemical Composition of Beef, Chicken and Pork Burgers

In order to better understand the composition of the meat used for burgers, the chemical composition was evaluated as listed in Table 1. It can be seen that the results of the analysis are consistent with the data in the literature [16,34].

The type of meat significantly influenced the chemical characteristics (*p* < 0.05). In particular, the PH showed significantly higher values in DM (35.53% ± 1.17), fat (9.68% ± 0.39) and protein (23.97% ± 0.55) than CH and BH (*p* < 0.05). The CH, on the other hand, had a lower fat content and a higher ash value (*p* < 0.05). It is known that cooking determines structural and compositional changes that influence the biological activities of the meat [35,36]. As expected, the cooked burgers had a chemical composition consistent with that found in raw meat; in fact, no major component losses were detected, and the significance of the differences between factor levels was consistent with that found in raw meat samples, as reported by other studies [12].

### 3.2. Antioxidant Activity

The presence and content of endogenous antioxidant components differed considerably due to the meat, breed, type of muscle considered, animal diet and farming system, significantly influencing both the nutraceutical value of the meat and the shelf life of the product [13,37]. In this study, the antioxidant activity of raw, cooked and digested beef, pork and chicken burgers was evaluated using spectrophotometric analyses of ABTS, FRAP and free thiols (Figure 1 and Figure 2). Given the complex reactivity of bioactive compounds and the lack of specific official methods, it was necessary to use different methods for the determination of the antioxidant properties of foods [9,38]. The ABTS assay measures the radical scavenging activity and the FRAP assay measures the iron-reducing potential of a sample, while the thiol assay measures the number of thiol groups (-SH), as well as glutathione and thiol groups in proteins, which play an essential role as antioxidants [39,40]. These compounds can act as free radical scavengers and metal ion chelators. Overall, the raw meat showed antioxidant activity (Figure 1 and Figure 2), a positive characteristic that enhances the qualitative aspects of the meat. Furthermore, significant differences in antioxidant activities were observed between burgers of different meats (Figure 1 and Figure 2). The antioxidant capacity of meat is the result of the ratio between endogenous antioxidant components and pro-oxidants. The endogenous antioxidant systems included all the non-enzymatic hydrophilic and lipophilic compounds, such as vitamins C and E, as well as proteins and peptides with an abundant presence of histidine, such as carnosine and anserine, but also endogenous enzymes such as superoxide dismutase (SOD), catalase (CAT) and glutathione peroxidase (GPx) [41]. Some authors [21,25] reported that meat proteins and peptides have the ability to scavenge free radicals and chelated metals; furthermore, the presence of –SH groups in amino acids and proteolysis by endogenous enzymes led to the formation of several peptides that can exhibit reactive thiol groups.

In the comparison between the raw burgers, beef showed higher FRAP and content of thiol groups (97.28 µmolTE/g of meat and 2.28 µmolSH/g of meat, respectively; *p* < 0.05), while chicken showed higher antioxidant activity as assessed by the ABTS test (*p* < 0.05). The ABTS and FRAP of raw burger samples were in line with the results reported by authors who found that the antioxidant-reductive capacity of the raw samples was lower than the scavenging capacity of ABTS^•+^ [41]. Furthermore, they found low FRAP values which could be related to the poor ability of meat antioxidants to reduce the ferric ion to its ferrous form.

The differences found in the meat of the different animals could therefore be due both to the composition and amino acid sequence of the proteins and to a different presence in the post mortem phase of calpain and cathepsin [42] responsible for the release of peptides containing Cys and Met residues on their nucleophilic side chains, with an electron donating effect. These may interact with free radicals to reduce and/or block the free radical chain reaction.

Cooking resulted in a decrease in the ABTS, FRAP and thiol values of the burgers (Figure 1 and Figure 2). Simonetti et al. [16] reported that heat treatment causes a destruction of the muscle cell structure, the inactivation of enzymatic and non-enzymatic antioxidants and the release of protein-bound iron, resulting in increased lipid and protein oxidation. The cooking process, in fact, triggers the generation of reactive oxygen species (ROS) [40], which leads to the consumption of antioxidant substances and a consequent decrease in the total antioxidant capacity. In particular, a decrease in free thiol content was observed (Figure 2; *p* < 0.001), in agreement with data reported by other authors [11,43]. In support, Renerre et al. [44] associated the decrease in free thiols with the oxidation of free thiol groups, available in the cysteine residues located on the outer wall of the protein, unlike the cysteine residues present inside the chain, which are consequently protected from the attack of free radicals even during heating treatment for longer times. Furthermore, as suggested by Simonetti et al. [12], during the cooking of meat, losses of cooking juice are detected, which could affect the reduced thiol content in cooked meat. After the cooking treatment, BH showed a higher thiol content (0.87 nmolSH-groups/g ± 0.05; *p* < 0.001) than CH and PH (0.56 ± 0.04 and 0.59 ± 0.02 nmolSH-groups/g, respectively). The loss rate of -SH groups was higher in the CH and BH samples, corresponding to losses of 64% and 63%, respectively, while the cooked PH showed a lower loss of 52%, indicating higher oxidative stability. Considering the ABTS values, cooked CH had the highest values (274.66 ± 26.61 µmol TE/g meat; *p* < 0.001) compared with BH (208.03 ± 18.32 µmol TE/g meat) and PH (248.94 ± 8.14 µmol TE/g meat), although the last one showed a greater percentage decrease (48.64%). Compared with the FRAP assay, cooked BH had the highest values (42.61 ± 4.12 µmol TE/g meat; *p* < 0.001), followed by CH and PH (36.76 ± 3.37 and 32.84 ± 2.19 µmol TE/g meat, respectively). However, considering the percent change in the antioxidant activity before and after cooking, it emerged that CH underwent the greatest decrease in FRAP (58%), followed by BH (56%) and PH (44%). The digestion of cooked meat, carried out by gastrointestinal enzymes, increased the antioxidant capacity of the meat (*p* < 0.001). Protein digestion, in fact, releases several peptides, characterised by biological activities of interest to human health for their antimicrobial, antioxidant and antihypertensive effects, into the intestinal lumen [31,45]. Pepsin treatment causes an increase in the concentration of low molecular weight (<3 kDa) biopeptides and an increase in the exposure sites of the released –SH groups, affecting their antioxidant activity [23,45]. Thus, the cooked–digested sample had a higher radical scavenging capacity than the cooked sample and control. Indeed, for the ABTS test, the digested samples showed a 7.7-fold increase in BH and almost 5-fold in CH and PH, compared with cooked samples of each meat, while the increase was almost 3.2 times in beef and almost 5 times in chickens and 6 times in pork compared with cooked samples, for FRAP values.

In the comparison between meat types, the digested–cooked BH showed the significantly highest ABTS values (1606.93 ± 46.01 µmol TE/g meat; *p* < 0.001), while the lowest value was observed in PH (1206.68 ± 42.32 µmol TE/g meat). Compared with the FRAP assay, digested–cooked PH showed the highest value (191.5 ± 8.88 µmol TE/g meat; *p* < 0.001), while the lowest value was found in digested BH (136.18 ± 4.97 µmol TE/g meat).

The digestion of cooked meat with gastrointestinal enzymes increased the thiol content (*p* < 0.001). Digested–cooked PHs showed a higher thiol content (191.5 ± 8.88 nmol of SH groups/g; *p* < 0.001) than digested–cooked CHs (179.51 ± 5.41 nmol of SH groups/g) and digested–cooked BHs (136.18 ± 4.97 nmol of SH groups/g).

In particular, the content of free -SH increased approximately 1.4-fold in the digested–cooked pork burger, 1.3-fold in digested–cooked beef burger and approximately 1.6-fold in digested–cooked chicken burger, showing increased susceptibility to protease action. Many authors [24,40,43] have correlated the initial oxidation of proteins upon digestion, suggesting that even partial oxidation will affect the unfolding of the protein structure, amplifying the susceptibility of proteins to protease action. Furthermore, some researchers [24,40] reported that increased chain unfolding, resulting in aggregation, will occur in a highly oxidative environment with the subsequent modification of protease active sites and reduced proteolytic susceptibility.

The digestion of meat leads to the release of Cys-containing peptides like glutathione [24], which are more stable in the gastrointestinal tract than free amino acids because the –SH groups are not oxidised and Cys residues are still active even at intestinal pH.

Some authors [24,40,43] have highlighted how the oxidation process contributes to the formation of disulphide and dityrosine bonds, with the consequent induction of aggregation of proteins that precipitate. Finally, the poor protein digestion ability would have a negative effect on human health due to non-hydrolysed proteins undergoing fermentation, damaging the colonic flora into phenol and p-cresol, which are mutagenic products, increasing the risk of colon cancer [46].

## 4. Conclusions

The aim of our work was to evaluate and compare, in vitro, the antioxidant activity of raw, cooked and cooked–digested pork, beef and chicken burgers. In general, beef, chicken and pork burgers could be considered as potential functional foods as they offer a rich source of bioactive components. The changes in the physicochemical state of the proteins during cooking influenced the antioxidant capacity of the meat by detecting a decrease in the values of ABTS, FRAP and the content of free thiols. In vitro gastrointestinal digestion improved the biological activity of the meat, which showed increased antioxidant activity. The type of meat influenced the chemical composition and biological capacity of the burgers. Beef burgers, before and after the cooking process, showed a higher thiol content and, consequently, a greater oxidative stability of proteins than chicken and pork burgers. After heat treatment, CH underwent the greatest decrease in FRAP (58%), while the greatest decrease in ABTS was observed in PH. In vitro gastrointestinal digestion improved the nutraceutical quality of the beef burger, which showed higher ABTS and thiol values, while the pork burgers showed higher FRAP values. Although meat is associated with non-communicable diseases and it is recommended to reduce its intake, the results obtained highlight the positive aspects of moderate meat consumption, which could bring benefits to the human body.

## Figures and Tables

**Figure 1 foods-12-04100-f001:**
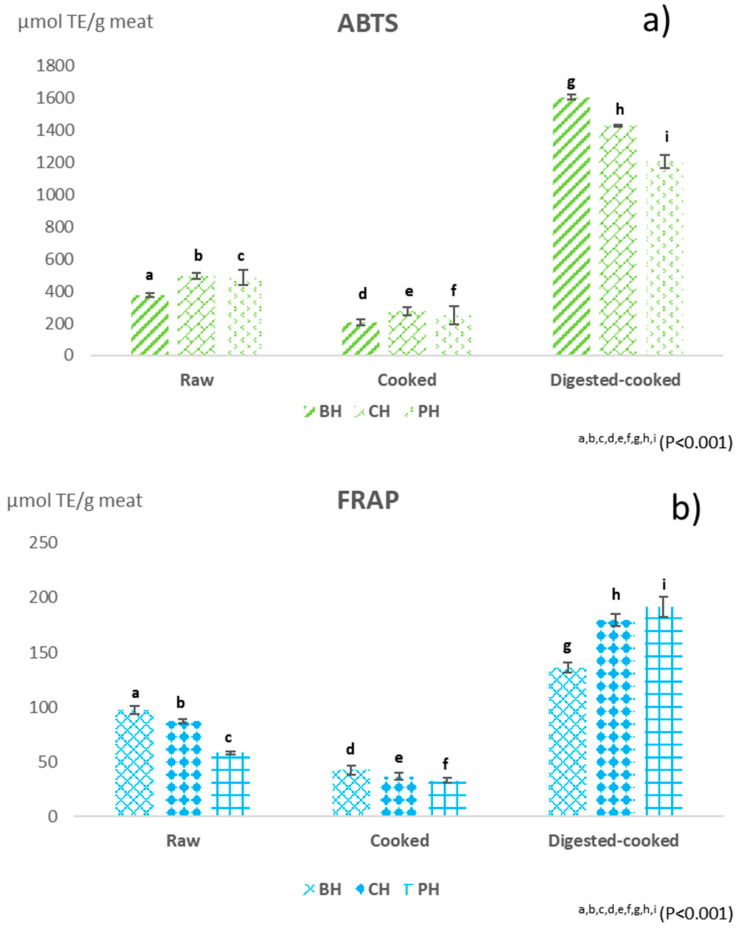
Antioxidant activity (ABTS and FRAP; µmolTE/g meat; *p* < 0.001; graphs: (**a**,**b**)) of burgers of different meats (beef (BH), chicken (CH), and pork (PH)) raw, cooked and digested–cooked. Note: ^a–i^: different letters are statistically different (*p* < 0.001); error bars indicate standard deviation.

**Figure 2 foods-12-04100-f002:**
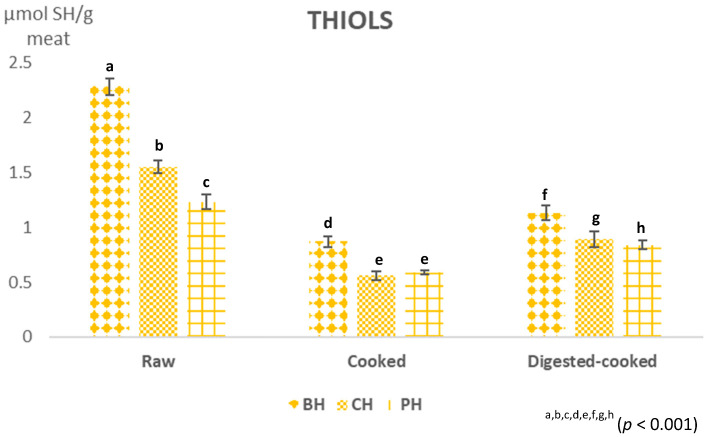
Thiols group content (µmolSH/g meat; *p* < 0.001) of burgers of different meats (beef (BH), chicken (CH) and pork (PH)) raw, cooked and digested–cooked. Note: ^a–h^: different letters are statistically different (*p* < 0.001); error bars indicate standard deviation.

**Table 1 foods-12-04100-t001:** Chemical composition, in percentages, of raw and cooked burgers distinct by meat (beef, chicken and pork).

		BH		CH		PH	
		µ	δ	µ	δ	µ	δ
DM	Raw	28.49 ^a^	1.04	28.20 ^a^	1.07	35.53 ^b^	1.17
	Cooked	39.06 ^c^	1.37	34.02 ^d^	1.20	43.99 ^e^	1.97
Fat	Raw	4.57 ^a^	0.15	3.46 ^b^	0.15	9.68 ^c^	0.39
	Cooked	6.01 ^d^	0.27	4.23 ^e^	0.21	12.09 ^f^	0.56
Protein	Raw	22.46 ^a^	0.96	23.09 ^b^	0.83	23.97 ^c^	0.55
	Cooked	30.73 ^d^	1.40	28.04 ^e^	0.69	29.72 ^f^	1.03
Ash	Raw	1.28 ^a^	0.07	1.40 ^b^	0.04	1.25 ^c^	0.12
	Cooked	1.92 ^d^	0.04	1.69 ^e^	0.09	2.16 ^f^	0.17

μ = means; δ = standard deviation. ^a–c^ means in the same row with different letters are statistically different based on Student’s *t*-test (*p* < 0.001). ^a–f^ means in the same column, for each parameter, with different letters are statistically different based on Student’s *t*-test (*p* < 0.001). DM: dry matter; BH: beef burger; CH: chicken burger; PH: pork burger

## Data Availability

The data presented in this study are available on request from the corresponding author.

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
