# Peer review of "Antioxidant Activity of Beef, Pork and Chicken Burgers before and after Cooking and after In Vitro Intestinal Digestion"

_foods, 2023, doi:10.3390/foods12224100_

Round 1

Reviewer 1 Report

Comments and Suggestions for Authors

The manuscript describes the comparative analysis on chemical composition and antioxidant activities among hamburgers prepared from beef, pork and chickens. Although the title has drawn attention, several points in the study strongly need further clarification. Since there are no line numbers in the manuscript, I'll try my best to explain those issues.

The context in the abstract was too broad. It doesn't reflect the results in the experiment. When comparing the results; higher ABTS, high FRAP, etc.,, the investigators did not state which samples were compared against each other. Therefore, several statements, in abstract and results & discussion, were unclear.

- The rationale and objective of the study were unclear, particularly why hamburger was used as a food model. Introduction was too general and did't emphasize the importance of this work. 

- M&M strongly needs a revision. Experimental design showing all factors and treatments should be clearly stated.

Section 2.1 :: Please indicate the composition of burgers; ratio of lean meat vs fat;  which meat was used for chicken, beef and pork; why the investigators choose not to control fat composition; what size of the prepared burgers (dimension & weight); how the burgers were stored (packed in any type of storage materials?) -- how did the investigators ensured that they were exposed to oxygen similarly as this could cause oxidation of the burger and affect antioxidant activities tested in the next experiment. Also, if only 6 burgers from each species were prepared and cooked -- did the raw burger samples prepared at the same time with the cooked ones? How many of the raw samples were prepared (how many replicates)?

2.3.1 Please indicate the sampling protocol from the burger.

2.3.2 In my understanding, the investigators extracted antioxidants from the samples using distilled water. Is this because you focused only on the polar antioxidants in the meat? Either way, please include this assumption / hypothesis along with explanation in the introduction and discussion. This would strengthen your manuscript remarkably.

2.4 According to the text, statistical analysis gave me a profound concern. Based on my understanding upon reading to this point, there were two factors; animal species (3 species) and cooking (raw vs cook). However, the investigators indicated that they used for ANOVA and Student's t-test in the analysis. Above all, Student's t-test was used for comparison among 3 species in Table 1 (shown as footnote under Table 1) -- which seems to be every inappropriate. No statistics was analyzed between raw vs cook in this case. As for figure 1 and 2, I assume that one-way ANOVA was used as there was no indication. However, shouldn't one-way ANOVA be used when there were interaction between two factors (species x cooking) -- In this regard, please indicate p value of the interaction.

Results and discussion: I do not see the link between chemical composition results and the title of this study. 

Please be careful when state the comparisons. For example, the first paragraph in page 6, "raw beef burgers showed higher FRAP and thiols". -- This statement was true when compared those parameters among raw samples but was not when the "raw beef" was compared among all bars in the figures. The way the figures were prepared appeared to lead readers to compare in the latter case. 

Overall, statement in conclusion was quite weak. It was not supported by results as no appropriate statistical analysis. Also, one may argue that similar benefits could be obtained from plant protein intake. Hence, this must be carefully stated; perhaps, with the manuscript revision and inclusion of further discussion. Also, I'd recommend adding some scientific limitation in the discussion so that the readers see that this manuscript has been well-thought.

Comments on the Quality of English Language

Please revise "," to "." for decimal digits in Table 1.

Please carefully check spellings in figures . Too many typos that implies to me that the investigators might not care much when preparing this manuscript. 

Author Response

Materials and Methods

M&M strongly needs a revision. Experimental design showing all factors and treatments should be clearly stated.

Section 2.1.: Please indicate the composition of burgers; ratio of lean meat vs fat;  which meat was used for chicken, beef and pork; why the investigators choose not to control fat composition; what size of the prepared burgers (dimension & weight); how the burgers were stored (packed in any type of storage materials?) - how did the investigators ensured that they were exposed to oxygen similarly as this could cause oxidation of the burger and affect antioxidant activities tested in the next experiment. Also, if only 6 burgers from each species were prepared and cooked -- did the raw burger samples prepared at the same time with the cooked ones? How many of the raw samples were prepared (how many replicates)?

Author's answer:

I thank the reviewer for the comments. Below are the responses to the comments:

- no fat was added to the hamburger mixture of each species in question;
- the requested information regarding the meat used for each species has been added to the text (Section 2.1.): BH: fresh ribeye; PH: fresh pork loins; CH: breast and thigh meat in a 1:1 ratio;

- the composition of the fat was not considered useful for the purposes of the study;

- 100 g were used for each hamburger, the information was also added in the text;

- conservation and packaging: no packaging was used because immediately after grinding and shaping, the burgers were kept at 4°C for two hours and then cooked (Section 2.2) and subjected to analysis. Furthermore, the meat of each species was processed at the same time and under the same conditions. Finally, 10 burgers were prepared in total, of which 4 were raw and 6 were cooked, the pattern was the same for each species.

Reviewer 1

2.3.1 Please indicate the sampling protocol from the burger.

Author's answer:

The sampling protocol was added to research work (Section 2.3.1.).

Reviewer 1

2.3.2 In my understanding, the investigators extracted antioxidants from the samples using distilled water. Is this because you focused only on the polar antioxidants in the meat? Either way, please include this assumption / hypothesis along with explanation in the introduction and discussion. This would strengthen your manuscript remarkably.

Author's answer:

I thank the reviewer for the comments. Distilled water has been used for the numerous hydrophilic components that have antioxidant power, among other things this extraction technique has been used by numerous authors (Sacchetti et al. 2008, Simonetti et al., 2016, 2019).

Reviewer 1

2.4 According to the text, statistical analysis gave me a profound concern. Based on my understanding upon reading to this point, there were two factors; animal species (3 species) and cooking (raw vs cook). However, the investigators indicated that they used for ANOVA and Student's t-test in the analysis. Above all, Student's t-test was used for comparison among 3 species in Table 1 (shown as footnote under Table 1) -- which seems to be every inappropriate. No statistics was analyzed between raw vs cook in this case. As for figure 1 and 2, I assume that one-way ANOVA was used as there was no indication. However, shouldn't one-way ANOVA be used when there was interaction between two factors (species X cooking) -- In this regard, please indicate p value of the interaction.

Author's answer

We realize we have made a serious mistake and we apologize. The text has been corrected and the statistical design has been revised.

Reviewer 1

Results and discussion:

I do not see the link between chemical composition results and the title of this study. 

Please be careful when state the comparisons. For example, the first paragraph in page 6, "raw beef burgers showed higher FRAP and thiols". -- This statement was true when compared those parameters among raw samples but was not when the "raw beef" was compared among all bars in the figures. The way the figures were prepared appeared to lead readers to compare in the latter case. 

Overall, statement in conclusion was quite weak. It was not supported by results as no appropriate statistical analysis. Also, one may argue that similar benefits could be obtained from plant protein intake. Hence, this must be carefully stated; perhaps, with the manuscript revision and inclusion of further discussion. Also, I'd recommend adding some scientific limitation in the discussion so that the readers see that this manuscript has been well-thought.

Author's answer:

I thank the reviewer for the comments. The chemical composition was entered to fully understand the sample. It is now known that the species influences the chemical-physical composition but also the bioactive components. We have also improved the introduction to better understand and support the purpose of the work.

Reviewer 2 Report

Comments and Suggestions for Authors

This paper describes an interesting study about the antioxidant capacity of cooked beef, pork, chicken hamburger meat after in vitro gastro-intestinal digestion.

The title of the manuscript should be revised since the 3 antioxidant activity assays are in vitro, I think is better Antioxidant activities of cooked beef, pork, chicken hamburger meat after in vitro gastro-intestinal digestion

In vitro, use italics

Indicate the type of meat of each sample studied; example bovine muscle (Longissimus dorsi)

To make it clear in the text that the digested is the cooked one in:

2.3.3. Free thiol groups

A supernatant volume (raw, cooked, and digested)

Tabla 1: in the table should be indicated what it means: BH, CH, PH

Insert standard deviations in all graphs

Letters indicating statistical difference must be in the graph legend

Figure 1, must indicate graph A and Graph B

Use: µmol not umol

Figure 2: Raw not paw

In particular, a decrease in free thiol content was recorded (Figure 1-2; P<0.001): please correct

In all the text when presenting the values of the analyses they should be with the standard deviation.

Example: After the cooking treatment, BH showed higher thiol content (0.87 nmolSH-groups/g; P<0.001) than CH and PH (0.56 and 0.59 nmolSH-groups/g, respectively). The loss rate of -SH groups was higher in the CH and BH samples, corresponding to losses of 64 and 63%, respectively, while the cooked PH showed a lower loss of 52%, indicating higher oxidative stability. Considering the ABTS values, cooked CH had the highest values (274.66 umol TE/g meat; P<0.001) compared to BH (208.03 umol TE/g meat) and PH (248.94 umol TE/g meat) although the latter showed a greater percent-age decrease (48.64%). Compared to the FRAP assay, cooked BH had the highest values (42.61 umol TE/g meat; P<0.001), followed by CH and PH (36.76 and 32.84 umol TE/g meat, respectively)

For in vitro digestion, the data of a control must be presented in all the analyses performed. See example in Food Chemistry 413 (2023) 135648.

Improve the discussion on the enhancement of antioxidant capacity in digested meat.

Comments on the Quality of English Language

Good quality, consider a minor revision.

Author Response

Reviewer 2:

- Comments and Suggestions for Authors:

This paper describes an interesting study about the antioxidant capacity of cooked beef, pork, chicken hamburger meat after in vitro gastro-intestinal digestion. 

The title of the manuscript should be revised since the 3 antioxidant activity assays are in vitro, I think is better Antioxidant activities of cooked beef, pork, chicken hamburger meat after in vitro gastro-intestinal digestion.

Author's answer

I thank the reviewer for the comments. The title of manuscript was has been changed as suggested.

Materials and Methods

- Comments and Suggestions for Authors:

To make it clear in the text that the digested is the cooked one in:

2.3.3. Free thiol groups: A supernatant volume (raw, cooked, and digested).

Author's answer

I thank the reviewer for the comments. The protocol was modified as suggested (Section 2.3.3).

- Comments and Suggestions for Authors:

Table 1: in the table should be indicated what it means: BH, CH, PH.

Author's answer

I thank the reviewer for the comments. BH, CH, PH have been indicated correctly.

- Comments and Suggestions for Authors:

Figure 1, must indicate graph A and Graph B

Use: µmol not umol

Figure 2: Raw not paw

In particular, a decrease in free thiol content was recorded (Figure 1-2; P<0.001): please correct

Author's answer

I thank the reviewer for the attention. All was corrected in the manuscript.

- Comments and Suggestions for Authors:

In all the text when presenting the values of the analyses they should be with the standard deviation.

Author's answer

I thank the reviewer and the standard deviation was been added.

- Comments and Suggestions for Authors:

For in vitro digestion, the data of a control must be presented in all the analyses performed. See example in Food Chemistry 413 (2023) 135648.

Author's answer

I thank the reviewer for the suggestion. We did not consider it useful to include it because it was considered in the final results which are net of digestion control.

Round 2

Reviewer 1 Report

Comments and Suggestions for Authors

Thank you for providing clarification. Because, your responses and your revised manuscript did not have any highlight on the modification. I'd tried my best to find whether there are some revision -- which are only in the statistical analysis and introduction sections if I'm correct. 

About the burger composition, now I have a better understanding. However, the burger is a broad term in different countries and regions. I still strongly recommend to at least they composed of lean meat and salt. If possible, could you please add your response about the sample number in your manuscript.

"Finally, 10 burgers were prepared in total, of which 4 were raw and 6 were cooked, the pattern was the same for each species."

I'd strongly recommend to add error bars in the bar graphs. It's very difficult to me to find significant difference between those bars with very much the same height, for example, figure 1, ABST between raw CH vs raw PH, and Fig 2 between raw PH vs digested BH.

Author Response

Author's answer:

I'm sorry to read that there have been no significant changes. Many changes have been made in the introduction, results and discussions section, also following the suggestions of the reviewers. Furthermore, the statistics section was recognized as an error on our part and we proceeded to modify the statistical model as reported in the revised article.

In the Materials and Methods section, the type of meat used for each species was specified and also the quantity of salt used. The number of samples used for raw burgers has been added in the Materials and Methods section.

Error bar has been added to each graph.

Reviewer 2 Report

Comments and Suggestions for Authors

Review abstract: it presents some mistakes and incorrectly formulated sentences.

Examples:The aim of our work was to evaluate and compare the in vitro digestibility and biological activities (antioxidant capacity) of cooked pork, beef and chicken hamburgers subjected to the in vitro gastrointestinal digestive model.

 in the human orgasm ¿?

About Graphs:

Insert standard deviations in all graphs

Letters indicating statistical difference must be in the graph legend as well as p≤…

The shape of the letters a and b in graph 1 does not look good.

On the comment below, please explain how the results were obtained considering the 2 blanks performed.

For in vitro digestion, the data of a control must be presented in all the analyses performed. See example in Food Chemistry 413 (2023) 135648.

I thank the reviewer for the suggestion. We did not consider it useful to include it because it was considered in the final results which are net of digestion control.

Simultaneously, a blank test was simulated in order to evalu-ate the possible impact of digestive enzymes on the planned analyses. The mixture was made up of water (instead of the sample) and gastrointestinal juices and enzymes. Furthermore, digestion of the different types of meat in the absence of digestive enzymes were also performed in order to distinguish the effects due to the presence of enzymes from those caused by the chemical conditions in the test.

Comments on the Quality of English Language

To make a general review.

Author Response

Author's answer

Review abstract: The wording was read again and worded correctly.

About Graphs: The graphs have been modified, adding error bars and reporting all the necessary information in the caption.

On the comment below, please explain how the results were obtained considering the 2 blanks performed.

The indicated manuscript used a gastrointestinal simulation following the INFOGEST protocol. The protocol we used was different (pH, duration of each step, enzyme-substrate ratio...) in terms of approach and time for each gastrointestinal phase. Our interest was to determine the antioxidant activities of the raw, cooked and cooked digested sample of each species. For each analysis, blank served to delete out interferences. The increase/decrease in the activities studied (ABTS, FRAP and Thiols) were compared between raw and cooked, cooked and cooked-digested, in order to highlight the increase in the antioxidant activities studied.